# Growing Human Hepatocellular Tumors Undergo a Global Metabolic Reprogramming

**DOI:** 10.3390/cancers13081980

**Published:** 2021-04-20

**Authors:** Fangrong Zhang, Yingchao Wang, Geng Chen, Zhenli Li, Xiaohua Xing, Csilla Putz-Bankuti, Rudolf E. Stauber, Xiaolong Liu, Tobias Madl

**Affiliations:** 1Gottfried Schatz Research Center, Molecular Biology and Biochemistry, Medical University of Graz, Neue Stiftingtalstraße 6/6, 8010 Graz, Austria; fangrong.zhang@medunigraz.at; 2BioTechMed-Graz, Mozartgasse 12/II, 8010 Graz, Austria; 3The United Innovation of Mengchao Hepatobiliary Technology Key Laboratory of Fujian Province, Mengchao Hepatobiliary Hospital of Fujian Medical University, Fuzhou 350025, China; yingchaowang@fzu.edu.cn (Y.W.); thestaroceanster@bjmu.edu.cn (G.C.); 11218083@zju.edu.cn (Z.L.); xingxiaohua2006@fjmu.edu.cn (X.X.); 4Department of Internal Medicine, Division of Gastroenterology and Hepatology, Medical University of Graz, Auenbruggerplatz 15, 8036 Graz, Austria; ordination@pro-endo.at (C.P.-B.); rudolf.stauber@medunigraz.at (R.E.S.); 5Xiamen Institute of Rare Earth Materials, Fujian Institute of Research on the Structure of Matter, Chinese Academy of Sciences, Xiamen 361024, China

**Keywords:** hepatocellular carcinoma, metabolomics, proteomics, NMR spectroscopy, predictive model

## Abstract

**Simple Summary:**

Metabolic reprogramming is a hallmark of malignancy. Hepatocellular carcinoma (HCC) cancer cells alterations in metabolism are due to the adaption to hypoxia and hypo-nutrient conditions. Several proteins and metabolites associated with glycolysis, tricarboxylic acid cycle and pyrimidine synthesis were found to be differentially regulated in serum, tumor and peritumoral tissues with increased tumor size, suggesting that microenvironment and tumor cell cooperate to regulate metabolism. In this study, the metabolomic characterization of HCC using paired tumor and adjacent liver tissues indicated tumor size-dependent metabolic reprogramming. Targeting cancer metabolism provides potential diagnostic and prognostic metabolic biomarkers. Our study brings new insight into the potential therapeutic use of metabolic targets and a methodological framework and diagnostic and prognostic metabolic markers that may be used in a clinical setting. The stratification of future clinical trials based on these metabolic subsets should improve the development of effective therapies and more intensive surveillance.

**Abstract:**

Hepatocellular carcinoma (HCC) is a common malignancy with poor prognosis, high morbidity and mortality concerning with lack of effective diagnosis and high postoperative recurrence. Similar with other cancers, HCC cancer cells have to alter their metabolism to adapt to the changing requirements imposed by the environment of the growing tumor. In less vascularized regions of tumor, cancer cells experience hypoxia and nutrient starvation. Here, we show that HCC undergoes a global metabolic reprogramming during tumor growth. A combined proteomics and metabolomics analysis of paired peritumoral and tumor tissues from 200 HCC patients revealed liver-specific metabolic reprogramming and metabolic alterations with increasing tumor sizes. Several proteins and metabolites associated with glycolysis, the tricarboxylic acid cycle and pyrimidine synthesis were found to be differentially regulated in serum, tumor and peritumoral tissue with increased tumor sizes. Several prognostic metabolite biomarkers involved in HCC metabolic reprogramming were identified and integrated with clinical and pathological data. We built and validated this combined model to discriminate against patients with different recurrence risks. An integrated and comprehensive metabolomic analysis of HCC is provided by our present work. Metabolomic alterations associated with the advanced stage of the disease and poor clinical outcomes, were revealed. Targeting cancer metabolism may deliver effective therapies for HCC.

## 1. Introduction

Hepatocellular carcinoma (HCC) is one of the most prevalent cancers around the world and the second-most common cause of cancer-related death [1]. HCC is a pathology processed with multiple etiologies, which are dynamic and multi-staged [2]. The observed variations in the age-, sex- and race-specific distributions of HCC in geographic regions are related to different abundance of risk factors, with chronic hepatitis B virus (HBV), chronic hepatitis C virus (HCV) infection and cirrhosis being the largest risk factors [3].

The high lethality (10–20/100,000, in East Asia) associated with HCC is mainly due to the late diagnosis and high post-operative recurrence, which is, in turn, due to the lack of symptoms at the early stage [4]. Currently, there is no standard or routine screening test for liver cancer. Depending on the local guidelines, ultrasound, X-ray Computed Tomography scan and α-fetoprotein (AFP) are the typical tests being used to screen for liver cancer [5,6,7]. The available tests have several limitations: for example, early/small tumors are difficult to be detected by ultrasound and computed tomography scan and often show negative AFP levels [8]. In contrast, AFP levels are often elevated under certain conditions, including pregnancy, hepatitis, cirrhosis and other types of cancer, causing up to 20% false-positive results [7,8]. Furthermore, there is currently no consensus regarding the risk stratification. These limitations complicate the recurrence surveillance of high-risk patients. Recently, several models to evaluate postoperative recurrence have been developed—for example, the Korean model, the Singapore Liver Cancer Recurrence (SLICER) score and the Surgery-Specific Cancer of the Liver Italian Program (SS-CLIP) [9,10,11]. However, the performance of these models is still not satisfactory for clinical practice, due to the lack of specific biomarker integration. Thus, novel biomarker candidates for the early diagnosis of HCC and new approaches that allow clinicians to estimate the risk of recurrence in an individual patient are urgently required.

Recent studies based on mass spectrometry and next-generation sequencing unveiled the activation status of signaling pathways and reprogramming of liver-specific metabolism in HBV-related HCC on the genomic and the proteomic levels [12,13]. MS-based proteomics can provide measures of the global changes in protein abundance related to the deregulation of signaling and metabolic pathways in HCC. Nonetheless, how the cancer metabolic phenotypes are driven by proteomic alterations remains unexplained in HBV-related HCC. Moreover, the application of proteomics-based biomarkers for diagnosis and prognosis remains difficult in a clinical setting [14,15]. One of the reasons of these complications might be imposed by the observation that tumors can undergo re-programming during growth to allow the tumor to adapt to changing conditions and challenges, including, for example, limited access to nutrients and oxygen [16,17]. However, how proteomic and metabolomic signatures change upon tumor growth in HCC remains unknown.

Nuclear Magnetic Resonance (NMR) spectroscopy-based metabolomics can provide an untargeted, quantitative snapshot of global metabolite abundance to provide additional biological insights, which cannot be deciphered by proteomics alone [18]. Compared to other “omics”, the metabolome provides the most direct snapshot of the actual functional and physiological state of biological networks [19]. Related to HCC, metabolic fingerprints have the potential to capture metabolic changes, which could help clarifying the pathogenesis and changes in environmental or lifestyle factors [20]. Untargeted metabolomics has been established as key technique for investigation of metabolic alterations in carcinogenesis [19,21], and the first metabolites have been identified to be changed in HCC [22,23]. Metabolomic research associated with HCC allows an unparalleled opportunity to discover metabolites for early diagnosis candidates and to assess the progression of treatment. The combination with MS-based proteomics promises to fill the current knowledge gap between HCC cancer proteomic and metabolomic phenotypes and the underlying molecular mechanisms [24].

Here, we performed a metabolomic analysis of patient serum samples and integrative analyses of metabolomic and proteomic [15] data from tumor and matched peritumoral liver tissues. Significant alterations in crucial metabolic and signaling pathways were revealed by this analysis. Strikingly, a comparison of the proteomic and metabolomic signatures obtained for different tumor sizes and their matched peritumoral tissues showed that both tumors and their surrounding tissues undergo proteomic and metabolic reprogramming during growth. Proteomic clustering resulted in a set of proteins that are differentially regulated in different-sized tumors. Several proteins, and metabolites involved in glycolysis, the tricarboxylic acid (TCA) cycle and pyrimidine synthesis were found to be differentially regulated with increased tumor sizes in serum, tumor and peritumoral tissues.

We further used our metabolomic data to develop a statistical model that can predict the risk of HCC recurrence based on clinical pathological data, metabolomics data and a combination thereof. This model will be valuable in predicting the clinical outcomes of the treatment, guiding follow-up surveillance or in the design of post-resection clinical strategies aiming to decrease the risk of recurrence. In conclusion, our study provides high-quality and high-content proteometabolomic resources of HBV-related HCC complementary to the sequencing-based data. Moreover, it highlights the therapeutic and prognostic implications and inherent regulatory mechanisms of metabolomic data benefiting clinical practice.

## 2. Materials and Methods

### 2.1. Patients

Tissues (carcinoma tissues (CT) and peritumoral tissues (PT)) and fasting sera were collected from 200 patients (male/female, 160/40) with HCC ranged from age 23 to 77 years. Several tumorous factors were collected, including tumor size, microvascular invasion, tumor lesion number, differentiation grade and portal vein tumor thrombus. According to the Barcelona Clinic Liver Cancer (BCLC) staging classification system [25] or TNM-based staging system [26], the tumor size is an important factor to grade HCC patients. In a number of more recent studies, the tumor size reflects changes in tumor growth, which were related with metastasis, microvascular invasion, operative complications and recurrence. Herein, serum, CT and PT were divided into 4 groups according to the tumor size (diameter ≤ 3 cm, grade I, *n* = 50; 3 cm < diameter ≤5 cm, grade II, *n* = 50; 5 cm < diameter ≤ 10 cm, grade III, *n* = 50; diameter > 10 cm, grade IV, *n* = 50). Serum and tissues were taken from each individual, i.e., serum, CT and PT were from the same patient. Intact encapsulation or distinct boundary tumors of patients through computed tomography scan were selected as the clinical samples. Of these, 45 of grade I patients, 40 of grade II, 48 of grade III and 41 of grade IV HCC patients were HBsAg-positive (Hepatitis B surface antigen) and one grade III patient with unknown HBsAg (all HCV (Hepatitis C virus) negative). All patients received the standard radical resection without any other therapies before surgery. The detailed clinical information is summarized in Appendix A. 

In addition, 42 HCC patients with cirrhosis and 23 cirrhosis controls without HCC enrolled at the Medical University of Graz were included. These patients were enrolled 2007–2009 in a biomarker study, and plasma samples stored at −70 °C were used for metabolomic analyses. In the HCC patients (male/female, 35/7; age range 54–83 years) all had underlying cirrhosis of different etiologies (alcohol, 13; NASH, 13; HCV, 12; other, 4). Cirrhosis controls without HCC were matched to age, sex and etiology.

This study was approved by the ethics committee of the Institution Review Board of Mengchao Hepatobiliary Hospital of Fujian Medical University, China (ethical code: 2018_067_01) and by the Ethics Committee of the Medical University of Graz (ethical code: 33-040 ex 20/21), respectively. Written informed consent was obtained from all patients.

### 2.2. MS Sample Preparation, Data Acquisition

In our previous study [15], CT and PT tissues were collected from 60 patients and were divided into 4 groups according to their tumor sizes, including small (*n* = 15), medium (*n* = 15), large (*n* = 15) and huge HCC groups (*n* = 15), respectively. The criteria for grouping was the same as in this study. After sample preparation, 5 individual samples with equal amounts from the same group were pooled together and were labeled using the iTRAQ kit, resulting in 3 biological repeats of each group, named as CT/PTI1, CT/PTI2, CT/PTI3, CT/PTII1, CT/PTII2, CT/PTII3, CT/PTIII1, CT/PTIII2, CT/PTIII3, CT/PTIV1, CT/PTIV2 and CT/PTIV3, respectively. Proteomic profiles of pooled samples were identified by 2D LC-MS/MS. Here, the raw data obtained in the previous study was used as input for the analysis.

### 2.3. NMR Sample Preparation, Data Acquisition and Analysis

Tissues samples were flash-frozen in liquid nitrogen and stored at −80°C until analysis. In each patient, approximately 9-mm^3^ CT and PT was resected and 200-µL serum was used for the metabolomics analysis. Serum and tissue sample preparation was conducted as described previously [27,28]. Tissuelyser-24 was used for homogenization at 60 Hz for 180 s (Lixin Co., Ltd, Shanghai, China). NMR measurements for ^1^H NMR metabolic profiling and analyses were performed as described and using a Bruker Avance III HD 600-MHz NMR spectrometer equipped with a TXI probe head [28,29]. Chenomx NMR suite 8.4 and reference compounds were used to identify the metabolites in the serum and tissue during the analysis. A Receiver Operating Characteristic (ROC) analysis was done in MetaboAnalyst 5.0 to evaluate the specificity and sensitivity of each metabolite. GraphPad Prism 5.01 (GraphPad Software, La Jolla, CA, USA) was employed to perform a univariate statistical analysis where the data are represented as the mean +/− standard deviation (SD).

### 2.4. Multivariable Prediction Model 

The model building and evaluation were performed by SPSS statistical software (V19.0, SPSS Inc., Chicago, IL, USA) or in R version 3.2.5 (R Foundation for Statistical Computing, Vienna, Austria). Two models to predict the recurrence of HCC were built using the derivation cohort. The difference between two models is whether the metabolites as a parameter are included in the model. Cox regression was conducted to evaluate the correlation between the clinicopathological parameters and metabolites. Univariable factors with *p* < 0.05 were incorporated into the multivariable cox regression analysis. Multivariate cox regression was then applied to establish the final model. As shown in Table 1 and Appendix A, two novel risk score formulae were developed, including or omitting metabolites, respectively. The risk score of enrolled HCC patients was calculated according to the aforementioned risk scoring formulae. The median risk score was subsequently used as cut-off to divide the patients into low- and high-risk groups. The survival differences between low- and high-risk groups were examined by a Kaplan–Meier analysis.

The sensitivity and specificity of this prognosis prediction model were assessed by the area under the time-dependent receiver operating characteristic (tdROC) curve (tdAUC), which was analyzed using the “tdROC” package of R software (4.0.3, R core Team, R Foundation for Statistical Computing, Vienna, Austria). The discriminatory performance of the models was assessed by Harrell’s c-index, Gönen & Heller’s K, as previously described [30,31,32].

## 3. Results

### 3.1. Metabolic Serum Profiles Change during Tumor Progression

Given that metabolic reprogramming is a hallmark of cancer development [33], we hypothesized that tumors of HCC patients undergo metabolic changes upon growth. To this end, we carried out an in-depth analysis using recently published proteomic data [15]. Proteomic clustering was performed based on differentially expressed proteins in different-size tumors, revealing 509 dysregulated proteins during tumor growth (Figure 1A). The proteins whose expression were altered the most significantly (*p* < 0.01), along with the increase in tumor size, are listed in Figure 1A. Among these differentially expressed proteins, many proteins have been demonstrated to be involved in glycometabolism, such as glyceraldehyde 3-phosphate dehydrogenase (GAPDH) [34] and phosphoglycerate kinase 1 (PGK1) [35], lipid metabolism such as apolipoprotein A4 (APOA4) [36] and protein synthesis or degradation such as plasminogen protein (PLG) [37] in cancer cells. Next, a Gene Ontology (GO) enrichment analysis identified the top terms (ranked by *p*-values) specific for the upregulated biological processes involving metabolic, mRNA, cellular metabolic, RNA, organic substances and catabolic processes (Figure 1B, upper panel). Tumors with larger sizes showed a decrease of mRNA processing related to the carboxylic acid, oxoacid, organic acid, small molecule and monocarboxylic acid metabolic processes (Figure 1B, lower panel). These results clearly indicate that metabolism undergoes tremendous change in tumors with different sizes. Therefore, we hypothesized that the metabolomic profile might consistently change in tumors with different sizes. 

To obtain a comprehensive molecular understanding of the metabolic changes in HCC patients, tumor tissues, their paired peritumoral tissues and serum collected from 200 HCC patients were used for a metabolomic analysis based on stringent criteria (see Materials and Methods). The study was designed to obtain samples of 50 HCC patients for each of the four groups defined according to a classification based on tumor size [38,39]. To this end, the HCC patients were divided into four subtypes, including small HCC (diameter ≤ 3 cm, grade I), medium HCC (3 cm < diameter ≤ 5 cm, grade II), large HCC (5 cm < diameter ≤ 10 cm, grade III) and huge HCC (diameter > 10 cm, grade IV).

Using untargeted NMR spectroscopy, we determined the metabolic fingerprints of the serum and tissues. A Principle Component Analysis (PCA) and Partial Least Squares-Discriminant Analysis (PLS-DA) of the serum samples showed that the serum metabolic profiles of HCC patients gradually change with increasing the tumor size (Figure 2A). While the metabolic profiles of HCC patients with small tumors are more homogenous, the profiles become more heterogeneous with the increasing tumor size. An inspection of the PCA loading plot revealed a strong increase of several ^1^H NMR signals between 3.5–4.0 ppm and between 5.4 and 5.7 ppm in a subset of patient samples with grade III and IV tumors. A statistical total correlation spectroscopy (STOCSY) [40] analysis confirmed that all the signals belong to the same group of metabolites (Appendix A). Although the corresponding metabolites could not be assigned unambiguously, we suspect that this group of metabolites corresponds to circulating DNA. This is in line with previous works reporting high levels of circulating tumor and cell-free DNA in HCC [41,42]. When comparing the differences in the metabolic fingerprints between the serum samples of patients with different tumor sizes, O-PLS-DA revealed increasing correlation coefficients R^2^Y up to 0.924 and a Q^2^ of 0.714 (*p* < 0.01) (Figure 2B,C) with increased clustering of the patient samples. In patients with different tumor sizes, altered serum metabolites were indicated by the reduced NMR spectra (Figure 2D) and indicated that the serum levels of glucose, pyruvate, glutamine, succinate, valine and 3-hydroxyisovalerate significantly changed (Figure 2E). The results obtained for succinate and valine were in line with a recent study using gas chromatography-mass spectrometry (GC-MS) on a small HCC patient cohort [43] and indicated that metabolites are stable markers for HCC. In line, we also observed significantly decreased glutamine and increased 3-hydroxyisovalerate in plasma samples of HCC patients with cirrhosis compared to matched cirrhosis controls without HCC in the European cohort (Graz, Austria) (Appendix A). We further analyzed HCC metabolomic results in subgroups (Asian vs. non-Asian and nonalcoholic steatohepatitis (NASH) vs. other etiologies). When comparing the metabolic fingerprints between serum/plasma from Asian (Chinese cohort) and non-Asian (European cohort), the O-PLS-DA revealed distinct clustering with correlation coefficients R^2^Y of up to 0.977 (*p* < 0.01) and a positive Q^2^ of 0.942 (*p* < 0.01) (Appendix A). We found significantly increased levels of branched-chain amino acids (BCAAs; valine, leucine and isoleucine) in the Asian group (Appendix A), which demonstrated changes in the metabolites may be relevant to HCC in a population-specific manner. Lower valine levels in the non-Asian group are in agreement with a previous study [44]. However, we cannot exclude the possibility that pre-analytics (plasma vs. serum), diet or lifestyle cause the differences. NASH is a common preneoplastic state of HCC [45]. Emerging data has revealed excessive hepatic lipid accumulation as the major contributor for NASH that sensitizes the liver to oxidative stress, along with subsequent necroinflammation [46,47]. The distinguishing clustering of plasma samples (European cohort) from NASH and other etiologies is shown in the score and validation plots of the O-PLS-DA (Appendix A) with a correlation coefficients R^2^Y value of 0.936 (*p* = 0.81) and the Q^2^ value of 0.217 (*p* = 0.02). We observed elevated concentrations of glucose in the NASH group (Appendix A). Consistently, a global meta-analysis demonstrated that NASH patients have a higher prevalence of type 2 diabetes and obesity compared to patients having nonalcoholic fatty liver diseases [48]. Glucose metabolism is complexly associated with other metabolic pathways (i.e., fatty acids metabolism), which will require further studies to obtain a deeper understanding [47].

### 3.2. Metabolic Profiles of Tumor and Peritumoral Tissues Change during Tumor Progression

In line with the serum data, the PCA and PLS-DA of tumor and peritumoral tissue samples showed that the metabolic profiles of tumor and peritumoral tissues gradually changed with the increasing tumor size, with grade III and IV tumor tissues showing the largest metabolic heterogeneity (Figure 3A). Interestingly, about half of the peritumoral tissue samples show a metabolic profile more similar to cancer tissues, whereas the other half showed a distinct metabolic phenotype. We hypothesized that this could be due to increased metastasis in these patients. To test our hypothesis, we performed a PCA analysis dividing the NMR metabolomics datasets into three groups: (i) cancer tissue, (ii) peritumoral tissue without metastasis and (iii) peritumoral tissue with metastasis. In agreement with our hypothesis, peritumoral tissue samples without metastasis clustered with a distinct metabolic phenotype, whereas peritumoral tissue samples with metastasis showed a more cancer-like metabolic phenotype (Appendix A). Reduced NMR spectra revealed glucose and succinate as the most significantly changed metabolites (Appendix A). Given that many peritumoral tissue samples showed a cancer-like phenotype even without reported metastasis, metabolic biomarkers might help to detect metabolic reprogramming and early metastasis in peritumoral tissue in the future. O-PLS-DA of pairs of tumor–tumor, peritumoral–peritumoral and tumor–peritumoral tissue revealed an increased clustering of patient samples with different tumor sizes (Figure 3B,C and Figure 4A,B,D,E). Strikingly, not only the metabolic profiles of tumors but, also, peritumoral liver tissues are affected, depending on the size of the tumors. Reduced NMR spectra revealed altered metabolites under all conditions compared, i.e., between tumor and peritumoral tissue (Figure 3D), as well as between different grade tumors (Appendix A) and peritumoral tissues (Appendix A), respectively. The levels of glucose, lactate; succinate; 3-hydroxyisovalerate; glutamate; choline; fumarate; nucleotides (uridine and UDP sugars) and other amino acids (isoleucine, leucine, valine, phenylalanine, tyrosine and lysine) were changed in tumor tissues compared to peritumoral tissue (Figure 3E), with a largest difference observed for grade IV tumor–peritumoral tissue pairs (Figure 3C). Within the tumor, the levels of glucose; lactate; glutamine; fumarate; nucleotides (uridine and UDP sugars) and other amino acids (isoleucine, leucine, valine, phenylalanine, tyrosine and lysine) were affected (Figure 4F). Interestingly, most of these metabolites were affected similarly in peritumoral tissues—in particular, lactate; glutamine; nucleotides (uridine and UDP sugars) and several amino acids (isoleucine, leucine, valine and lysine) (Figure 4C). Glucose, lactate and leucine have been identified previously by solid-state ^1^H NMR in HCC tumor tissues and are in line with our results [23]. In summary, our results indicate that NMR metabolomics are well-suited to studying tumor metabolism and that metabolism of the tumor is strongly coupled to its environment (Figure 4C).

### 3.3. Metabolite Panels Enable Diagnosis and Prognosis Potential of HCC

We performed a supervised analysis to identify robust and representative metabolites for the serum-based discrimination against the tumor size and prognosis. In the serum, increased levels of succinate and decreased levels of glutamine and pyruvate were observed depending on the tumor grade (Figure 2E). To assess the specificity and sensitivity for each serum metabolite, an ROC analysis was applied. The highest predictivity associate with a tumor grade, as indicated by the highest values of the Area under the Curve (AUC), was obtained for pyruvate (AUC, 0.849), followed by glutamine (AUC, 0.84), and succinate (AUC, 0.809) (Appendix A). Glutamine also exhibited a high diagnostic accuracy for the discrimination of HCC with underlying cirrhosis from cirrhosis without HCC in the European cohort (AUC, 0.787) (Appendix A). These metabolites also showed an elevated response probability to the prognosis, which was consistent with the above order for pyruvate (AUC, 0.625) followed by glutamine (AUC, 0.621) and succinate (AUC, 0.583) (Appendix A). Using a combination of these metabolites yielded a slightly higher AUC (~0.65) compared to the AUC using single metabolites (pyruvate of serum, 0~0.625), revealing an improved predictive estimate of recurrence (Appendix A).

Surgical resection offers a potential curative treatment for HCC patients. Nonetheless, as a major cause of mortality, recurrence happens in around 50–70% of the patients within the 5 years’ recurrence period [49,50]. Out of 200 patients of the Chinese cohort tested in this study, a total of 96 patients (48%) developed recurrence within 5 years of surgery. 

We hypothesized that metabolites might assist to enable clinicians to determine the suitable therapy and surveillance of patients who were at higher risk of recurrence after surgery. Despite a lack of consensus regarding the optimal tool for stratification, Chan et al. recently proposed a prognostic model called the Early Recurrence After Surgery for Liver tumor (ERASL) [51] for the surgical treatment of the patients with HCC according to the accessible pathological and clinical parameters. As described before, the tumor number and size, the preoperative alpha-fetoprotein (AFP) level, the preoperative albumin-bilirubin (ALBI) grade and gender were first combined as independent predictors to first validate the pre-ERASL model. In our cohort, 86 out of 184 patients (46.73%) developed recurrence within 5 years of surgery. An additional 16 patients with missing data in our cohort were excluded; in eight patients, no AFP could be detected, in six patients, no exact AFP value could be reported and, in two patients, no ALBI could be determined). We found that the ERASL model is suitable to predict the 5-year (preoperative) recurrence of HCC patients with our own data provided (*p* = 0.03) (Appendix A).

Among the 22 clinicopathological parameters analyzed, a univariate Cox regression analysis was performed, and tumor number, tumor size, TNM stage (8th version [26]), serum alanine transaminase (ALT), alkaline phosphatase (ALP), serum glutamic-oxaloacetic transaminase (AST), Gamma-glutamyltransferase (γ-GT) and tumor encapsulation were found to be potentially relevant with *p* < 0.05 (Appendix A). Here, in contrast to previous studies [52], we did not find microvascular invasion to improve the prediction of recurrence-free survival in our derivation cohort (100 randomly picked samples, *p* = 0.377), although it could improve the prediction in the entire cohort (200 samples, *p* = 0.0041). Patients were divided randomly into two groups of 100 patients each to generate a derivation and validation cohort, and the eight relevant parameters were used to build a modified and slightly improved preoperative model to predict the 5-year recurrence (*p* = 0.006 for the derivation and *p* = 0.025 for the validation cohort) (Figure 5A,B). The formula of the constructed preoperative model is shown in Appendix A. Using 0.691 as the cut-off value for the clinicopathological preoperative model score, two prognostically distinct groups were stratified, a low risk (5-year recurrence-free survival, RFS = 64/37%) and a high-risk (5-year RFS = 33/0%) group in the derivation and validation groups, respectively.

Next, we aimed to validate the suitability of metabolites to improve the preoperative model by using the same derivation and validation cohorts. To this end, all significantly changed serum and tissue metabolites and their ratios were subjected to a univariable Cox regression analysis. Several metabolites were found to be potentially relevant, with *p* < 0.05 in the univariable Cox regression analysis: serum succinate and glucose, CT lactate, PT lactate, isoleucine, glucose, UDP sugars and valine, as well as the ratios of serum pyruvate/PT valine, serum pyruvate/PT isoleucine, serum pyruvate/CT lactate, serum pyruvate/PT leucine and serum succinate/pyruvate (Appendix A). These 13 metabolite parameters were combined with the eight relevant clinicopathological parameters to build an integrated metaboclinicopathological model. The combination of parameters showed a comparable performance in the derivation group (from *p* = 0.006 to *p* = 0.010) and an improved performance in the validation group (from *p* = 0.025 to *p* = 0.001) (Figure 5C,D). The multivariable analysis identified the tumor number, alanine aminotransferase, succinate and pyruvate as the key parameters related to a poor prognosis. Using these variables, a metaboclinicopathological preoperative model was constructed, and the independent parameters and their formulae are shown in Table 1. Using 0.874 as the cut-off value, two prognostically distinct groups were stratified, a low-risk (5-year RFS = 61/50%) and high-risk (5-year RFS = 42/6%), in the derivation and validation groups, respectively. The discriminatory performance of the models was compared via Harrell’s c-index, Gönen & Heller’s K and tdAUC, as shown in Table 2. By including NMR metabolite data, the metaboclinicopathological preoperative model showed a better discriminatory performance compared to the clinicopathological preoperative model. Both models may help to design clinical guidelines trials.

## 4. Discussion

A comprehensive global genomic, transcriptomic, proteomic and phosphoproteomic analysis of HCC provided the first insights into the underlying molecular mechanisms and first insights into the biological understanding of HCC [12,53]. Herein, the global metabolomics data provided new insights into the biological understanding of HCC, with particular implications related to the clinical and therapeutic understanding of HCC. NMR spectroscopy is a powerful tool in this regard, as it is characterized by its high reproducibility and outstanding suitability for in vitro diagnostics (IVD). The integrated metabolomic characterization of the serum and paired tumor and peritumoral liver tissue tumor samples at different grades of tumor growth revealed metabolic reprogramming, communication between tumor and peritumoral tissue and clinically and therapeutically relevant metabolite markers in HCC. 

Our integrated analysis revealed alterations of the metabolic pathways in the serum, tumor tissue and peritumoral liver tissue. Moreover, we discovered that metastasis affects the metabolic profile of peritumoral tissue. The related metabolites can be used in follow-up studies as markers in a clinical setting for the early diagnosis of HCC and to detect metastasis in peritumoral tissue, which is a difficult task—in particular, for micrometastasis [54]. In addition, we compared the metabolic changes of serum/plasma in subgroups (Asian vs. non-Asian and others vs. NASH). Our results showed that BCAAs may act as population-specific HCC metabolites and identified glucose as a plasma biomarker candidate for distinguishing NASH from other etiologies. A limitation of our analysis concerns the inability to subdivide HBV and HCV in the same population. 

Strikingly, the metabolic phenotypes of the serum and tumor tissue, as well as peritumoral liver tissue, are shifted gradually with the increasing tumor size. Together with an alteration of the proteomics phenotype, this indicates that tumors undergo metabolic reprogramming with increasing size. Metabolic reprogramming is a hallmark of tumor growth, independent of their carcinogenetic origin [55]. Our results in the serum and tissues of HCC patients show that the tumor shifts metabolic pathways and indicates that the resulting change in the nutrient supply is indispensable to overcome nutrient starvation and the changing environmental conditions. Notably, metabolites related with glycolysis, the tricarboxylic acid (TCA) cycle and pyrimidine synthesis were changed in tumor tissues of different stages. The increasing demand of growing tumors for glucose and glutamine is well-visible in the serum and both tumor and peritumoral tissues, respectively. In line with this, the lactate levels increased both in tumor and peritumoral tissues, as well as the expression of glutamine transporters (i.e., SLC1A5) and glycolytic enzymes, such as glucose-6-phosphate isomerase (GPI), phosphoglycerate kinase (PGK1), aldolase A (ALDOA) and hexokinase 2 (HK2) (Figure 6). Despite the increased glutamine uptake, enzymes involved in the conversion of glutamate to α-ketoglutarate are downregulated, such as glutamate dehydrogenases (GLUD1), glutamic-oxaloacetic transaminases (GOT1) and glutamic-pyruvate transaminases (GPT). This might be coupled to the altered expression of downstream TCA enzymes. A prominent example is the drop in the tumor suppressor succinate dehydrogenase (SDHA and SDHB) with the increasing tumor size. Converting succinate to fumarate, the loss of this enzyme is in agreement with reduced levels of fumarate in tumors and increased levels of succinate in the serum. In addition to glutamine, other amino acids such as branched-chain amino acids (BCAAs; valine, leucine and isoleucine) can function as opportunistic fuel sources for cells. In line with this expression of related catabolic enzymes such as BCAT2 are increased in tumor tissues, although other enzymes are undetectable. 

Strikingly, we observed large changes in the metabolites associated with pyrimidine synthesis (UDP sugars and uridine), which are in line with the recent proteogenomic analysis of HCC [12] and the upregulation of CAD protein (CAD), dihydroorotate dehydrogenase (DHODH), uridine 5’-monophosphate synthase (UMPS), CTP synthase 2 (CTPS2) and nucleoside diphosphate kinase A (NME1). These findings suggested that pyrimidine synthesis pathways could be an alternative target for HCC therapy.

Encouraged by the large changes in HCC metabolic profiles, we validated and adapted a recently proposed a preoperative model that enables risk assessment of 5-year recurrence before resection for the inclusion of metabolomics data. Our validation shows that the recently proposed model can be applied to our setting and that it was capable of stratifying patients into two groups with discrete risk profiles. In the low-risk group comprising about 56.52% of patients among the entire cohort, only 34.61% developed 5-year recurrence, whereas in the intermediate-risk group of 42.39% patients, 61.53% developed 5-year recurrence (Appendix A). Although the metabolomics data on their own were not sufficient to predict the 5-year recurrence (*p* = 0.000 for the derivation and *p* = 0.705 for the validation cohort), the inclusion of metabolomics data improved the predictions of the clinical model. Currently, the serum AFP levels and ultrasonography are regarded as common means for the surveillance of HCC and the early detection of recurrence [56]. Ultrasonography shows a low level of sensitivity for the surveillance of HCC, especially in patients with cirrhosis [56]. Indeed, AFP leads to high rates of false negatives for HCC, both in the case of the Chinese (66/192) and the European cohorts (20/42). The ROC curve analysis and AUC revealed a higher diagnostic performance of our metaboclinicopathological model (AUC 0.669) than AFP (AUC 0.518) to predict the 5-year recurrence (Appendix A). Our models are clinically relevant, as they enable the identification of a small, although potentially manageable group of patients with a high risk of recurrence for which an adjuvant therapy and more intensive surveillance could be provided. The benefit of adding metabolomics data to the set of clinicopathological parameters should be further validated in a multicenter study in the future. 

In all, after the curative surgery for HCC, recurrence of the tumor is a common and potentially severe complication. Our combined clinicopathological and metabolomic model is a clinically relevant, validated and potent tool to predict the 5-year recurrence. Further prospective studies are needed to demonstrate the applicability of our model in patient allocation for adjuvant trails and more frequent follow-up.

## 5. Conclusions

An integrated and comprehensive metabolomic analysis of HCC is provided by our current work. It could be established that poor clinical outcomes, coupled with an advanced disease stage, were the key factors associated with metabolic alterations. Targeting cancer metabolism, especially purine metabolism, may offer a promising strategy for the effective treatment of HCC. The methodological framework, diagnostic and prognostic metabolic markers capable of being used in a clinical setting are provided by our study, besides generating a high-quality untargeted analysis of HCC metabolism, also benefitting the basic research.

## Figures and Tables

**Figure 1 cancers-13-01980-f001:**
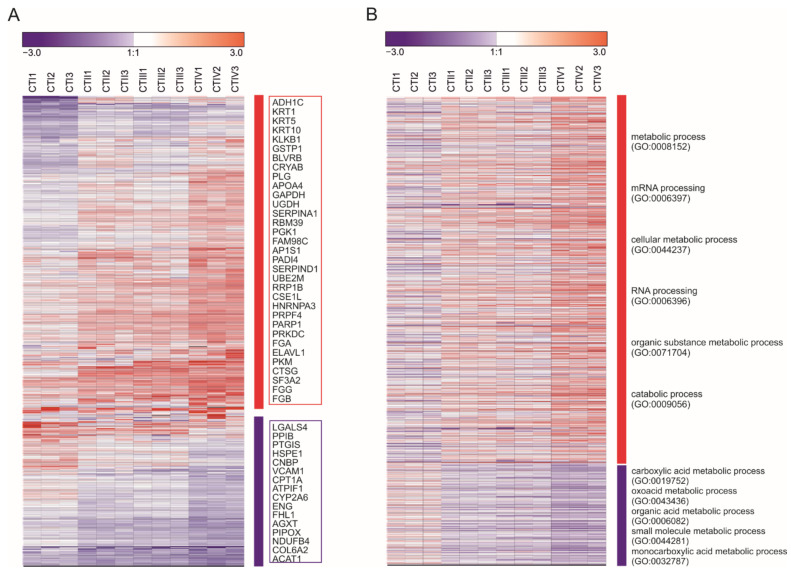
Proteome changes during tumor progression. (**A**) Hierarchal molecular clustering of 509 dysregulated (1.5-fold up or down regulation) proteins in tumors of different HCC subtypes. Each column represents a pool of patient samples, and rows indicate proteins. Up- and downregulated proteins were compared to the peritumoral tissue, respectively. The red frame surrounds the proteins whose upregulation was significantly associated with larger tumor sizes (*p* < 0.001), while the purple frame surrounds proteins whose downregulation was significantly associated with bigger tumor sizes (*p* < 0.001). (**B**) An enriched analysis of the Gene Ontology Biological Processes (GOBP) using protein panels based on upregulated and downregulated proteins to generate top-term significant biological process. Each column represents a pool of patient samples, and rows indicate proteins involved in that particular biological process. Cells are colored according to their Z-score (log2 of the relative protein abundance). The analysis was performed on the HCC cancer tissue proteome dataset from a previous work [15].

**Figure 2 cancers-13-01980-f002:**
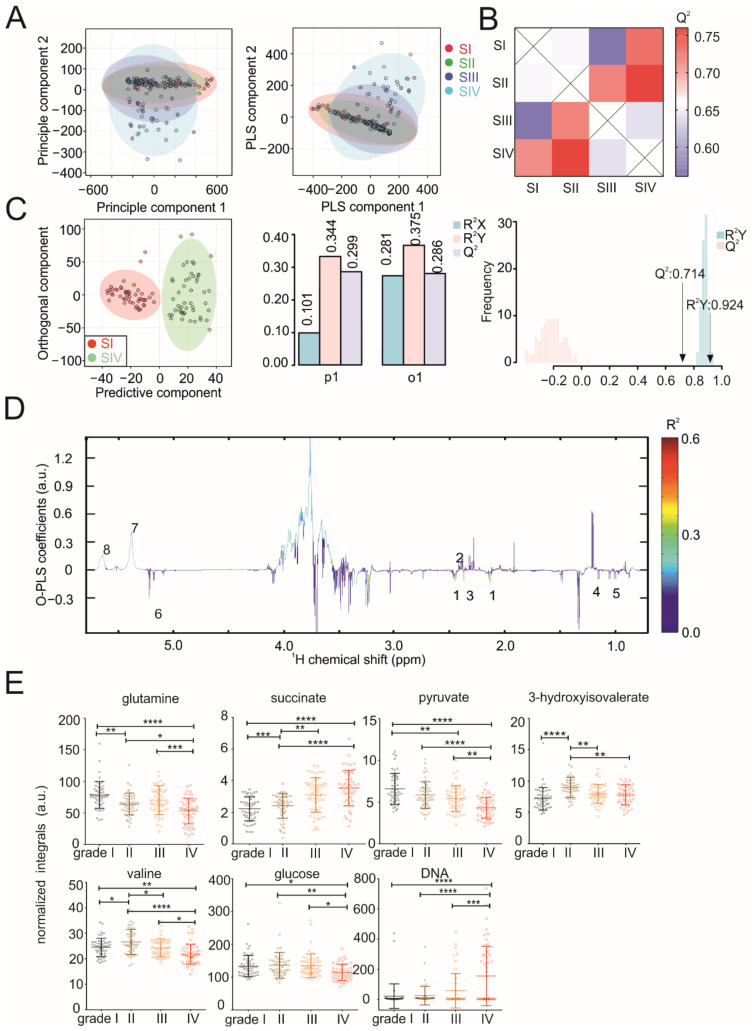
NMR metabolomics analysis of serum samples. (**A**) PCA and PLS-DA plots of serum samples. (**B**) Heatmap showing O-PLS-DA-derived Q^2^ for pairwise comparisons of the serum samples. (**C**) O-PLS-DA plot of the serum samples, including a cross-validation. (**D**) The reduced NMR spectrum reveals altered components in normalized serum samples. Positive covariance corresponds to the components present at increased concentrations, whereas negative covariance corresponds to decreased component concentrations. The predictivity of the model is represented by R^2^. (1) glutamine, (2) succinate, (3) pyruvate, (4) 3-hydroxyisovalerate, (5) valine, (6) glucose and (7) and (8) DNA. (**E**) Statistical analysis of individual metabolites in serum samples. Statistical differences among multiple groups (one-way ANOVA) are indicated by *p*-values of < 0.05 (*), < 0.01 (**), < 0.001 (***) or < 0.0001 (****).

**Figure 3 cancers-13-01980-f003:**
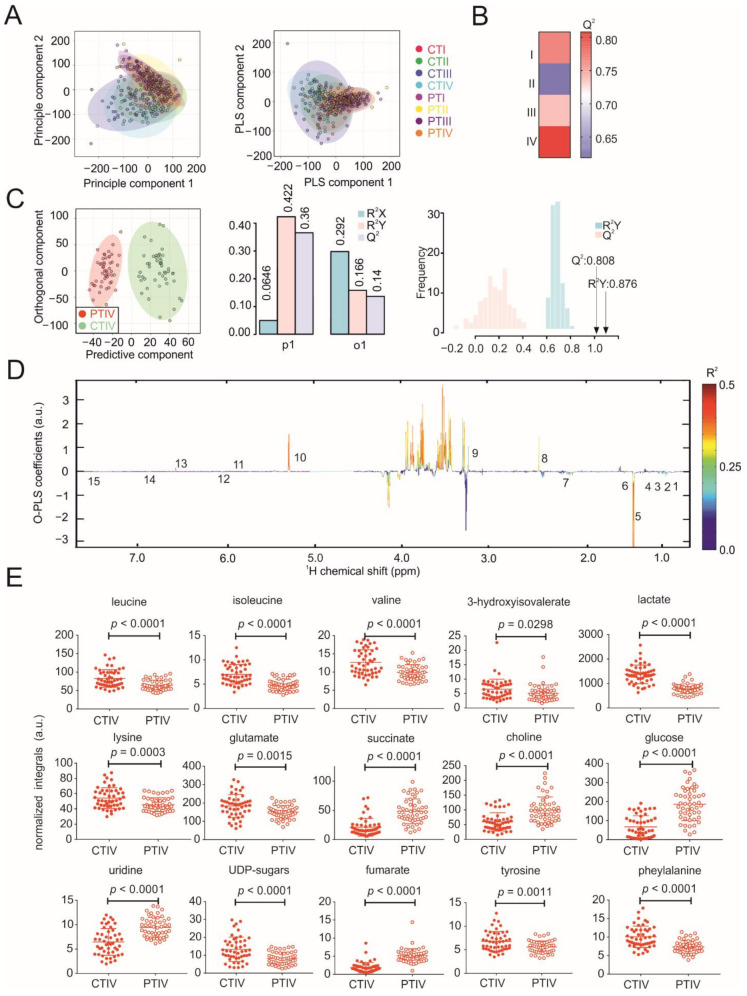
NMR metabolomics analysis of the tissue samples. (**A**) PCA and PLS-DA plots of tissue samples (grade IV carcinoma tissues (CTIV) and peritumoral tissues (PTIV)). (**B**) Heatmap showing O-PLS-DA-derived Q^2^ for pairwise comparisons of the tissue samples. (**C**) O-PLS-DA plot of the tissue samples, including a cross-validation. (**D**) The reduced NMR spectrum reveals altered components in the normalized tissue samples. Positive covariance corresponds to the components present in increased concentrations, whereas negative covariance corresponds to decreased component concentrations. The predictivity of the model is represented by R^2^. (1) leucine, (2) isoleucine, (3) valine, (4) 3-hydroxyisovalerate, (5) lactate, (6) lysine, (7) glutamate, (8) succinate, (9) choline, (10) glucose, (11) uridine, (12) UDP sugars, (13) fumarate, (14) tyrosine and (15) phenylalanine. (**E**) Statistical analysis of the individual metabolites in the tissue samples.

**Figure 4 cancers-13-01980-f004:**
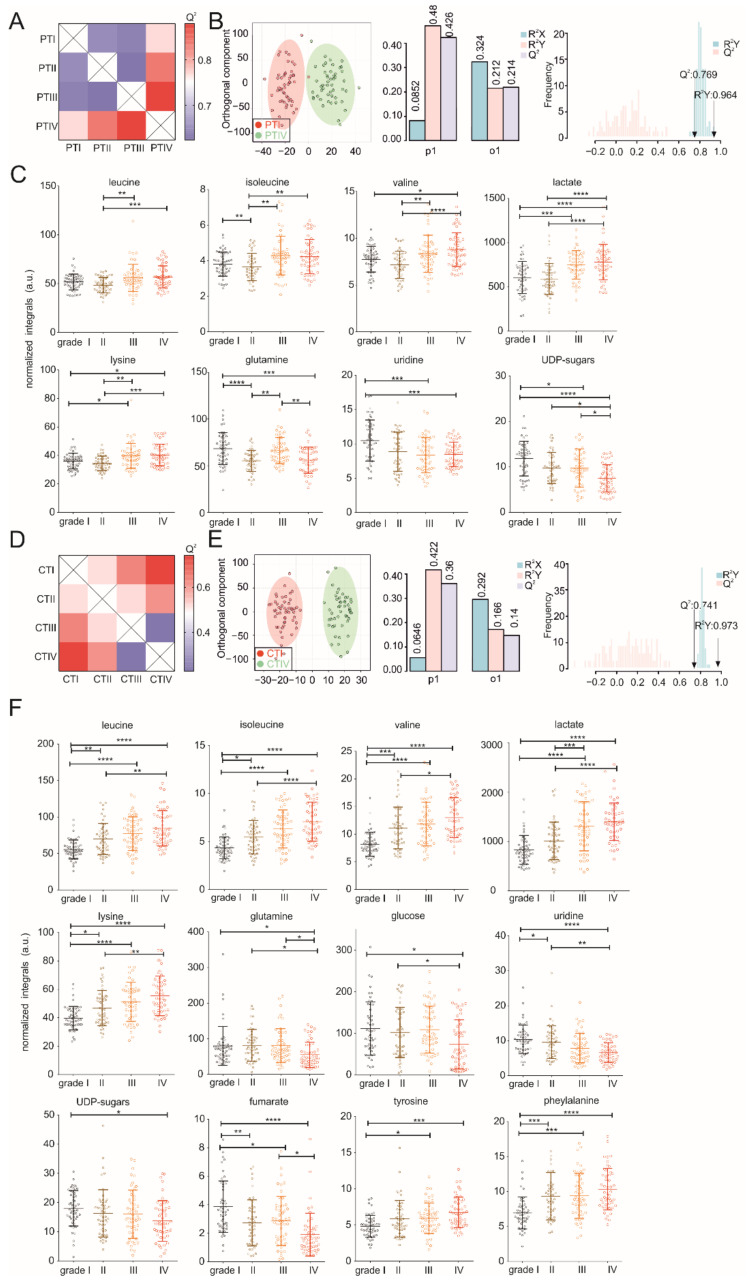
NMR metabolomics analysis of peritumoral and cancer tissue samples at different stages. (**A**) Heatmap showing O-PLS-DA-derived Q^2^ for pairwise comparisons of peritumoral tissue samples. (**B**) O-PLS-DA plot of peritumoral samples, including a cross-validation. (**C**) Statistical analysis of individual metabolites in peritumoral tissue samples. (**D**) Heatmap showing O-PLS-DA-derived Q^2^ for pairwise comparisons of cancer tissue samples. (**E**) O-PLS-DA plot of cancer tissue samples, including a cross-validation. (**F**) Statistical analysis of individual metabolites in cancer tissue samples. Statistical differences among multiple groups (one-way ANOVA) are indicated by *p*-values of < 0.05 (*), < 0.01 (**), < 0.001 (***) or < 0.0001 (****).

**Figure 5 cancers-13-01980-f005:**
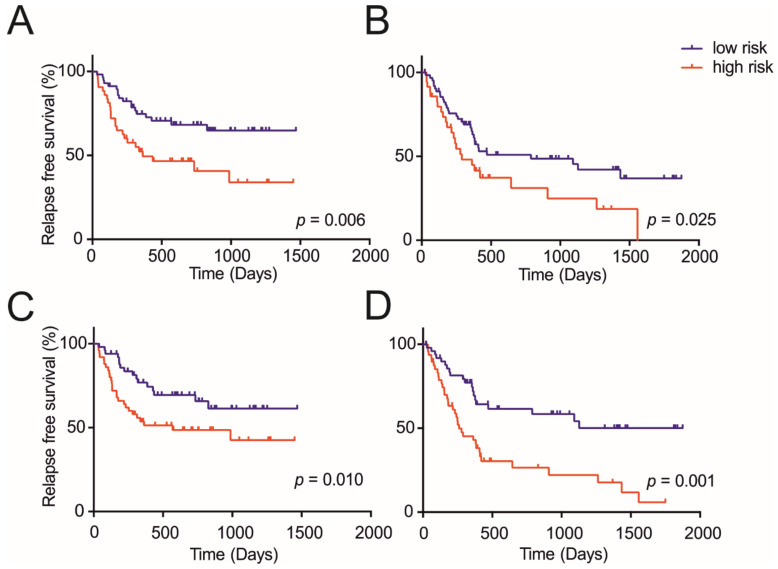
Recurrence-free survival (RFS) according to risk groups defined by the clinicopathological and metaboclinicopathological model. Clinicopathological model (**A**) derivation group and (**B**) validation group. Metaboclinicopathological model (**C**) derivation group and (**D**) validation group.

**Figure 6 cancers-13-01980-f006:**
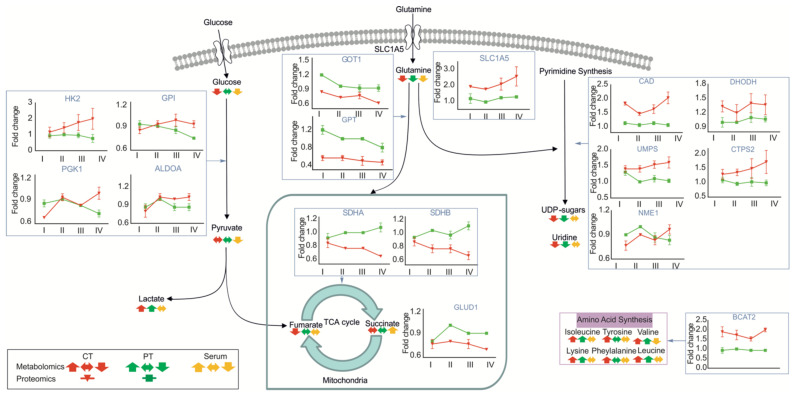
Schematic overview of growing human hepatocellular tumors undergoing a global metabolic reprogramming and proteomic changes. Metabolic genes and metabolites involved in glycolysis, the tricarboxylic acid (TCA) cycle and pyrimidine synthesis. Red line represents tumor tissues, and green line represents peritumoral tissues in proteomics. Red arrow represents tumor tissues, green arrow represents peritumoral tissues and yellow arrow represents the serum in the metabolism.

**Table 1 cancers-13-01980-t001:** Multivariable Cox regression analyses of the prognostic factors in the derivation and validation groups of the Chinese cohort with metabolic factors.

Variable	Derivation Group (*n* = 100)			Validation Group (*n* = 100)		
	Hazard Ratio (95% CI)	β-Estimate (95% CI)	*p*-Value	Hazard Ratio (95% CI)	β-Estimate (95% CI)	*p*-Value
ALT						
<50	Ref	Ref		Ref	Ref	
>50	2.557 (1.191, 5.490)	2.552 (1.291, 5.046)	0.007	0.916 (0.473, 1.774)	−0.088 (−0.749, 0.573)	0.794
serum succinate/serum pyruvate	4.572 (1.360,15.372)	1.520 (0.307, 2.733)	0.014	1.968 (1.185, 3.268)	0.677 (0.170, 1.184)	0.009
Tumor number	2.242 (1.540, 3.264)	0.807 (0.432, 1.183)	<0.0001	2.228 (1.614, 3.075)	0.801 (0.479, 1.123)	<0.0001

Model score = tumor number * 0.807 + serum succinate / serum pyruvate * 1.52 + ALT (0, <50; 1, ≥50) * 0.937; PI = e^ model score/(1 + e^model score); Cut-off to generate the risk groups: PI ≤ 0.874 (low) and PI > 0.874 (high). ALT, Alanine aminotransferase.

**Table 2 cancers-13-01980-t002:** Prognostic performance of the models.

Measure of Discrimination	With Metabolites		Without Metabolites	
	Derivation (SE)	Validation (SE)	Derivation (SE)	Validation (SE)
Harrell’s c-index	0.661 (0.047)	0.638 (0.041)	0.610 (0.045)	0.573 (0.040)
Gönen & Heller’s K	0.658 (0.041)	0.619 (0.036)	0.669 (0.056)	0.637 (0.052)
tdAUC (5 years)	0.671 (0.059)	0.667 (0.060)	0.618 (0.055)	0.572 (0.052)

Standard errors (SE) were estimated from 200 bootstrap samples; tdAUC, areas under the time-dependent receiver operating characteristic curve.

## Data Availability

The data presented in this study are available on request from the corresponding author.

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
