# Peer review of "Growing Human Hepatocellular Tumors Undergo a Global Metabolic Reprogramming"

_cancers, 2021, doi:10.3390/cancers13081980_

Round 1
Reviewer 1 Report
HCC still remains a high medical burden and novel options for diagnosis, surveillance and prediction of prognosis are urgently needed. The authors describe metabolic reporgramming in a series of human HCC samples. Alterations in tumor metabolism have been proposed more than 100 years ago by Warburg, but so far, no satisfying approach has been established in clinical settings. The authors use novel technologies like NMR and MS approaches and present a novel predictive scoring system.
Overall, such approaches are of interest to further and better characterize HCC. Like in other studies, sample size is often rather small, esp. when a breakdown to HCC subgroups (Asian vs non-Asian, HBV vs HCV vs other etiologies) is done. This should be discussed more critically. Of special interest would be a more thorough description of metabolic changes in NASH related HCC since here factors like FGF signaling, metabolic changes in lipid pathways etc are commonly found and are driving the inflammatory setting.
It would be good to show a direct comparison of the new prognostic score proposed by the authors to currently used clinical scores. This would allow a better judgement on clinical usability of such technically more advanced methods.
Overall, the manuscript is very well written and results are clearly presented.
Author Response
HCC still remains a high medical burden and novel options for diagnosis, surveillance and prediction of prognosis are urgently needed. The authors describe metabolic reprogramming in a series of human HCC samples. Alterations in tumor metabolism have been proposed more than 100 years ago by Warburg, but so far, no satisfying approach has been established in clinical settings. The authors use novel technologies like NMR and MS approaches and present a novel predictive scoring system.
We thank Reviewer 1 for her/his positive evaluation of our manuscript.
Overall, such approaches are of interest to further and better characterize HCC. Like in other studies, sample size is often rather small, esp. when a breakdown to HCC subgroups (Asian vs non-Asian, HBV vs HCV vs other etiologies) is done. This should be discussed more critically. Of special interest would be a more thorough description of metabolic changes in NASH related HCC since here factors like FGF signaling, metabolic changes in lipid pathways etc are commonly found and are driving the inflammatory setting.
We thank Reviewer 1 for her/his suggestion. As suggested, we further analyzed HCC metabolomics results in subgroups (Asian vs. non-Asian, NASH vs. other etiologies). When comparing the metabolic fingerprints between serum/plasma from Asian (Chinese cohort) and non-Asian (European cohort), the O-PLS-DA revealed distinct clustering with correlation coefficients R2Y of up to 0.977 (p < 0.01) and a positive Q2 of 0.942 (p < 0.01) (Figure S1C). We found significantly increased branched-chain amino acids (BCAAs; valine, leucine, and isoleucine) in the Asian group (Figure S1D), which demonstrated changes in metabolites may be relevant to HCC in a race-specific manner. Lower valine levels observed in the non-Asian group are in agreement with a previous study [1]. However, we cannot exclude the possibility that pre-analytics (plasma vs. serum), diet or lifestyle cause the differences. NASH is a common preneoplastic state of HCC [2]. Emerging data reveal excessive hepatic lipid accumulation as the major contributors for NASH that sensitized the liver to oxidative stress along with subsequent necroinflammation [3,4]. The distinguishing clustering of plasma samples (European cohort) from NASH and other etiologies is shown in the score and validation plots of the O-PLS-DA (Figure S1F) with correlation coefficients R2Y value of 0.936 (p=0.81) and the Q2 value of 0.217 (p=0.02). We observed elevated concentrations of glucose in the NASH group (Figure S1E). Consistently, a global meta-analysis demonstrated that NASH patients have a higher prevalence of type 2 diabetes and obesity compared to patients having nonalcoholic fatty liver diseases [5]. Glucose metabolism is complexly associated with other metabolic pathways (i.e. fatty acids metabolism), which will required further studies to obtain a deeper understanding [3]. All patients in the Asian group were HCV negative, and on the contrary, all patients in the non-Asian group were HBV negative. Thus, we were not able to make a reasonable comparison (HCV vs. HBV) using our data sets. We have included this limitation into our discussion: A limitation of our analysis concerns the inability to subdivide HBV and HCV in the same race.
Figure. S1. NMR metabolomics analysis of serum/plasma samples (C) O-PLS-DA plot of HCC serum/plasma samples, including cross validation (Asian vs. non-Asian). (D) Statistical analysis of individual metabolites in serum/plasma samples (Asian vs. non-Asian). (E) Statistical analysis of individual metabolites in serum/plasma samples (others vs. NASH). (F) O-PLS-DA plot of HCC serum/plasma samples (European cohort), including cross validation (others vs. NASH).
It would be good to show a direct comparison of the new prognostic score proposed by the authors to currently used clinical scores. This would allow a better judgement on clinical usability of such technically more advanced methods.
Overall, the manuscript is very well written and results are clearly presented.
We thank Reviewer 1 for her/his suggestion. As suggested, we have extended the discussion of the comparison of our prognostic score and currently used clinical scores. We have revised the manuscript accordingly.
Currently, serum AFP levels and ultrasonography are regarded as common means for the surveillance of HCC and the early detection of recurrence [6]. Ultrasonography shows a low level of sensitivity for the surveillance of HCC, especially in patients with cirrhosis [6]. Indeed, the AFP leads to high rates of false-negative for HCC, both in case of the Chinese (66/192) and the European cohorts (20/42). ROC curve analysis and AUC revealed a higher diagnostic performance of our metaboclinicopathological model (AUC 0.669) than AFP (AUC 0.518) to predict 5-year recurrence (Figure S4A).
Figure. S4. ROC analysis of AFP, metaboclinicopathological model and altered metabolites. (A) ROC curve of metaboclinicopathological model (high vs. low risk) and AFP (positive vs. negative) in HCC patients with recurrence versus non-recurrence.

Reviewer 2 Report
This manuscript reports a comprehensive metabolic analysis of serum samples, tumor and peritumoral tissues from the same HCC patients. Considered the high mortality and the high recurrence risk of HCC, the search for novel targets to develop novel diagnostic strategies or novel therapies is of great interest. Moreover, proteomic and metabolic profiling offer the opportunity to gain mechanistic hypotheses.
The paper is well written and clear, the only exception is the simple summary that seems to be too confounding. I think it should be rewritten at least partially (L22: are due; L25: tissues; L26-28: unclear; please specify better).
Minor points:
Title: undergoes
L102: Here,
L107: tisses
L112: tissues.
L124: peritumoral tissues
L124 fasting sera
L253-256: discussion material
L335-353: discussion material
Author Response
This manuscript reports a comprehensive metabolic analysis of serum samples, tumor and peritumoral tissues from the same HCC patients. Considered the high mortality and the high recurrence risk of HCC, the search for novel targets to develop novel diagnostic strategies or novel therapies is of great interest. Moreover, proteomic and metabolic profiling offer the opportunity to gain mechanistic hypotheses.
We thank the Reviewer 2 for her/his positive evaluation of our manuscript.
The paper is well written and clear, the only exception is the simple summary that seems to be too confounding. I think it should be rewritten at least partially (L22: are due; L25: tissues; L26-28: unclear; please specify better).
Minor points:
Title: undergoes (should be undergo as we wrote before)
L102: Here,
L107: tissues
L112: tissues.
L124: peritumoral tissues
L124 fasting sera
L253-256: discussion material
L335-353: discussion material
We thank Reviewer 2 for her/his suggestions and carefulness. We have corrected the typographical errors and revised the manuscript accordingly.
L26-28: In this study, metabolomic characterization of HCC using paired tumor and adjacent liver tissues indicated tumor size-dependent metabolic reprogramming. Targeting cancer metabolism provides potential diagnostic and prognostic metabolic biomarkers.
L253-256: The analysis was performed on the HCC cancer tissues proteome dataset from previous work [7].
L335-353: Figure 3. (A) PCA and PLS-DA plots of tissue samples [grade IV carcinoma tissues (CTIV) and peritumoral tissues (PTIV)].
References
- Di Poto, C.; He, S.; Varghese, R.S.; Zhao, Y.; Ferrarini, A.; Su, S.; Karabala, A.; Redi, M.; Mamo, H.; Rangnekar, A.S. Identification of race-associated metabolite biomarkers for hepatocellular carcinoma in patients with liver cirrhosis and hepatitis C virus infection. PLoS One 2018, 13, e0192748.
- Muir, K.; Hazim, A.; He, Y.; Peyressatre, M.; Kim, D.-Y.; Song, X.; Beretta, L. Proteomic and lipidomic signatures of lipid metabolism in NASH-associated hepatocellular carcinoma. Cancer research 2013, 73, 4722-4731.
- Nakagawa, H.; Hayata, Y.; Kawamura, S.; Yamada, T.; Fujiwara, N.; Koike, K. Lipid metabolic reprogramming in hepatocellular carcinoma. Cancers 2018, 10, 447.
- Anstee, Q.M.; Reeves, H.L.; Kotsiliti, E.; Govaere, O.; Heikenwalder, M. From NASH to HCC: current concepts and future challenges. Nature reviews Gastroenterology & hepatology 2019, 16, 411-428.
- Younossi, Z.M.; Koenig, A.B.; Abdelatif, D.; Fazel, Y.; Henry, L.; Wymer, M. Global epidemiology of nonalcoholic fatty liver disease—meta‐analytic assessment of prevalence, incidence, and outcomes. Hepatology 2016, 64, 73-84.
- Tzartzeva, K.; Obi, J.; Rich, N.E.; Parikh, N.D.; Marrero, J.A.; Yopp, A.; Waljee, A.K.; Singal, A.G. Surveillance imaging and alpha fetoprotein for early detection of hepatocellular carcinoma in patients with cirrhosis: a meta-analysis. Gastroenterology 2018, 154, 1706-1718. e1701.
- Wang, Y.; Liu, H.; Liang, D.; Huang, Y.; Zeng, Y.; Xing, X.; Xia, J.; Lin, M.; Han, X.; Liao, N.; et al. Reveal the molecular signatures of hepatocellular carcinoma with different sizes by iTRAQ based quantitative proteomics. J Proteomics 2017, 150, 230-241, doi:10.1016/j.jprot.2016.09.008.
